# EAD-Net: Efficiently Asymmetric Network for Semantic Labeling of High-Resolution Remote Sensing Images with Dynamic Routing Mechanism

Qiongqiong Hu [1], Feiting Wang [2] and Ying Li [1,*]

1   School of Computer Science, Northwestern Polytechnical University, Xi'an 710129, China;
    qionghu@mail.nwpu.edu.cn
2   Department of Computer Technology and Application, Qinghai University, Xi'ning 810016, China;
    wangfeiting187@163.com
*   Correspondence: lybyp@nwpu.edu.cn

**Abstract:** Semantic labeling of high-resolution remote sensing images (HRRSIs) holds a significant position in the remote sensing domain. Although numerous deep-learning-based segmentation models have enhanced segmentation precision, their complexity leads to a significant increase in parameters and computational requirements. While ensuring segmentation accuracy, it is also crucial to improve segmentation speed. To address this issue, we propose an efficient asymmetric deep learning network for HRRSIs, referred to as EAD-Net. First, EAD-Net employs ResNet50 as the backbone without pooling, instead of the RepVGG block, to extract rich semantic features while reducing model complexity. Second, a dynamic routing module is proposed in EAD-Net to adjust routing based on the pixel occupancy of small-scale objects. Concurrently, a channel attention mechanism is used to preserve their features even with minimal occupancy. Third, a novel asymmetric decoder is introduced, which uses convolutional operations while discarding skip connections. This not only effectively reduces redundant features but also allows using low-level image features to enhance EAD-Net's performance. Extensive experimental results on the ISPRS 2D semantic labeling challenge benchmark demonstrate that EAD-Net achieves state-of-the-art (SOTA) accuracy performance while reducing model complexity and inference time, while the mean Intersection over Union (mIoU) score reaching 87.38% and 93.10% in the Vaihingen and Potsdam datasets, respectively.

**Keywords:** deep learning; small-scale object; low-level features; channel attention mechanism; asymmetric decoder



## 1. Introduction

The exponential growth of remote sensing technology [1–4] has made high-quality remote sensing images (RSIs) acquisition more accessible and cost-affordable. High resolution is a crucial characteristic of RSIs, with spatial resolutions generally ranging between 5 and 10 cm. This level of detail unveils various objects' features, such as edges and textures, displaying them clearly. It thereby lays a solid foundation for subsequent research applications, including road monitoring [5], wildfire detection [6], building extraction [7,8], cloud detection [9], land-cover classification [10], and change detection [11,12]. Accurately extracting these features and effectively utilizing them has resulted in increased attention and rapid advancement in semantic labeling (also known as semantic segmentation in computer vision) for high-resolution remote sensing images (HRRSIs).

Before the advent of deep learning methods, traditional image segmentation techniques were prevalent. These methods relied on features, such as boundaries, color, and texture to divide images into regions, but these regions did not contain semantic information. Supervised learning methods can obtain highly accurate segmentation results

using a large amount of labeled data, especially if the training data are sufficient and well-represented. In recent years, unsupervised semantic segmentation methods [13,14] for RSIs can classify features without pre-labeled training data. These methods are very effective in dealing with large-scale remote sensing data because the methods do not require an expensive and time-consuming manual labeling process. However, unsupervised methods usually fall short in the accuracy of semantic segmentation compared to supervised learning methods, because they do not make use of accurate labeling information and require more tuning and optimization in the algorithm. Without labeling information to guide the learning process, unsupervised learning methods may be more susceptible to noise and outliers. Semi-supervised learning methods [15,16] are a learning strategy that combines supervised and unsupervised learning, which uses a small amount of unlabeled data and a large amount of unlabeled data for training, which can be utilized to improve the performance of the model, reducing the need for a large amount of labeled data, and saving labeling costs. However, if the labeled data are scarce, the semi-supervised learning method may not be able to fully utilize this limited information, resulting in poor performance. Meanwhile, the effectiveness of semi-supervised learning is affected by the quality and quantity of unlabeled data, and the performance is not as stable as supervised learning.

The advent of deep convolutional neural networks (DCNNs) [17] has proven to be successful in solving the semantic labeling of HRRSIs due to their ability to automatically learn complex, correlated contextual information end-to-end [18–20]. In 2014, Fully Convolutional Network (FCN) [21] was first successfully applied to the semantic segmentation task. Following this, numerous models with subtle network architectures were designed, such as U-Net [22], Conv-Deconv [23], and SegNet [24].

Building upon FCN and U-Net, numerous well-designed networks, including PSP-Net [25], DeepLabv3 [26], and DeepLabv3+ [27], have been proposed to enhance segmentation accuracy by fusing contextual information and expanding the receptive field through the use of pyramid pooling modules (PPM) or atrous spatial pyramid pooling (ASPP). Simultaneously, numerous networks [28–30] have been proposed by establishing pixel-level relationships with attention mechanisms to improve segmentation performance.

To reduce the computational power required by dense prediction, lightweight architectures have been proposed and applied to semantic segmentation, such as LR-ASPP [31], CGNet [32], BiSeNet [33], MobileNet [34], ESPNet [35], and RepVGG [36].

To minimize manual intervention in the design of network architectures, neural architecture search (NAS) [37,38] has emerged as a viable solution. NAS can automatically select and combine various neuron operations within a predefined search space to identify the optimal network architecture, yielding impressive results in segmentation [39]. This approach significantly streamlines the process of designing efficient and accurate network architectures, reducing the need for extensive manual labor and experimentation.

Manually designed static networks and those obtained through NAS-based searching share a common limitation: they attempt to encode all pixels within a single network framework, resulting in a lack of dynamic adaptability to various scale distributions in real-world scenarios. In contrast, dynamic routing selection [40] enables networks to generate forward propagations dynamically, allowing them to adapt their structures for feature encoding according to the input images. To address these issues, we propose an efficiently asymmetric network called EAD-Net, which utilizes a dynamic module for semantic labeling of HRRSIs. To reduce the complexity of the learning model and fully utilize the precise feature information contained in the labeled data, we use a supervised learning approach to improve the segmentation accuracy of the small-scale targets by dynamically selecting the different paths that can fully extract their features based on their pixel amount, while using the ASPP module to comprehensively consider the feature information of feature targets at different scales. The aim is to be able to improve the accuracy of small-scale feature targets without degrading the segmentation performance of other categories.

## 2. Proposed Method

The network structure, as shown in Figure 1. Unlike other segmentation models, EAD-Net's encoder does not employ skip connections, which helps reduce memory occupation during the training stage. Additionally, the network employs the RepVGG block in each layer to enhance feature extraction capabilities.

The feature enhancement module dynamically adjusts the network structure based on the proportion of small-scale objects, thus improving their segmentation accuracy—a challenging task that typically impacts overall performance. To further enhance feature representation and generation, EAD-Net introduces a channel attention module that assigns weights to channels according to their importance in the input image.

EAD-Net utilizes the ASPP semantic extraction module, which enables capturing multi-scale features through a multi-sore effective receptive field. The decoder employs $3 \times 3$ convolutions instead of direct skip connections, allowing for the full integration of low-level semantic segmentation while effectively reducing image redundancy.

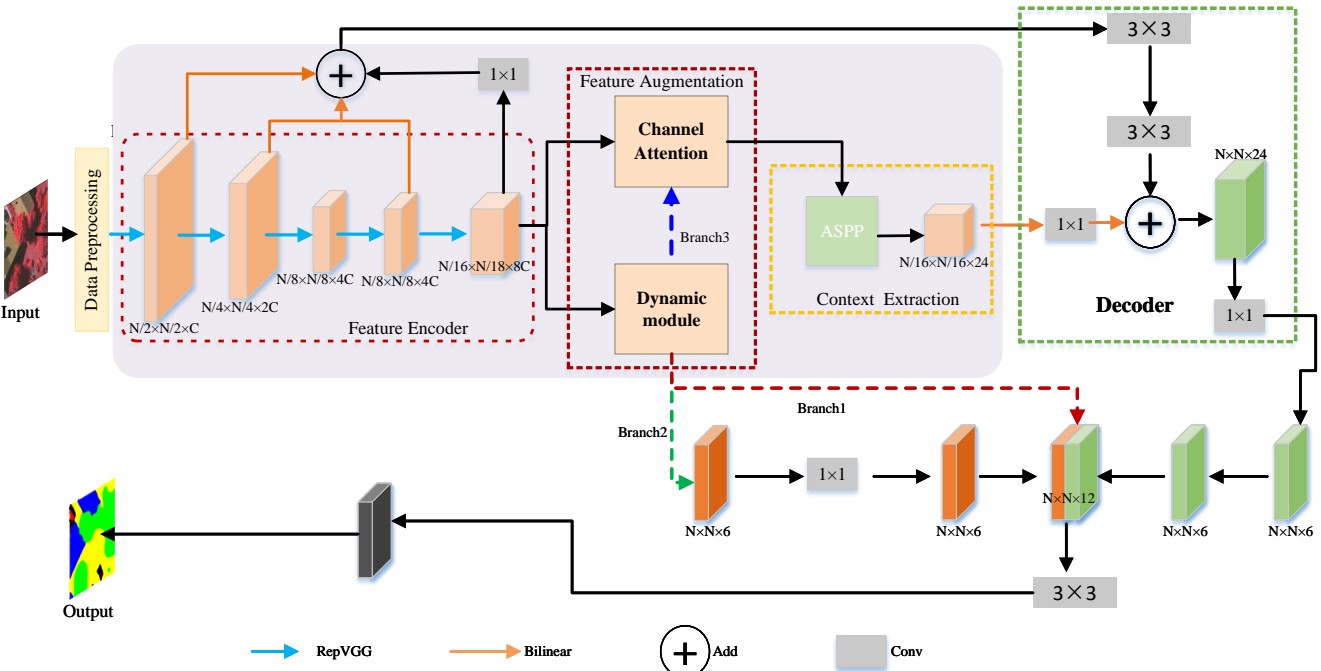

**Figure 1.** The structure of the EAD-Net.

### 2.1. Encoder Module

EAD-Net introduced the RepVGG block, shown in Figure 2, as a solution to this problem. The RepVGG block uses convolution (stride = 2) and regulation to replace the pooling operation, thus avoiding a reduction in spatial information. It also employs a parallel multi-branch structure, which includes multiple residual connections, to enhance the network's feature representation. The $3 \times 3$ convolution operation further accelerates the training process.

### 2.2. Feature Augmentation

To address the issue of misclassification caused by the sparse pixels of small-scale objects in HRRSIs segmentation tasks, a dynamic module has been proposed.

The dynamic module, as shown in Figure 3, utilizes a dynamic routing selection strategy that decides whether to use a particular feature map and which path to take based on the percentage of pixels belonging to small-scale objects in the input feature matrix.

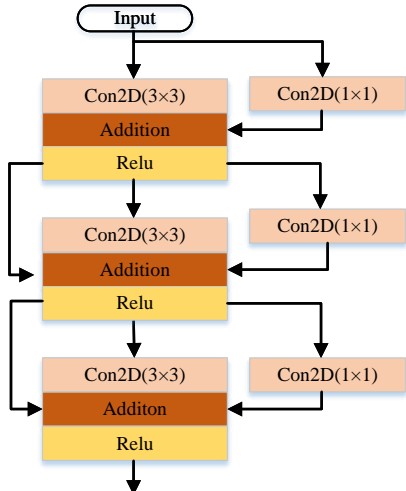

**Figure 2.** The architecture of the Re-Parameterization Visual Geometry Group block.

In this module, three threshold values are set at 15%, 10%, and 5%, respectively. These threshold values help to determine the extent of involvement of small-scale objects in the input image. When the percentage of pixels belonging to small-scale objects is below the threshold, the dynamic module utilizes the conventional feature maps for segmentation. However, when the percentage exceeds the threshold, the module activates the dynamic routing selection strategy.

The dynamic routing selection strategy involves the following steps:

1. Calculate the percentage of pixels belonging to small-scale objects in the input feature matrix.
2. Compare the calculated percentage with the predefined threshold values.
3. Utilize the appropriate feature maps and paths based on the activation of the dynamic module to improve misclassification caused by sparse pixels.

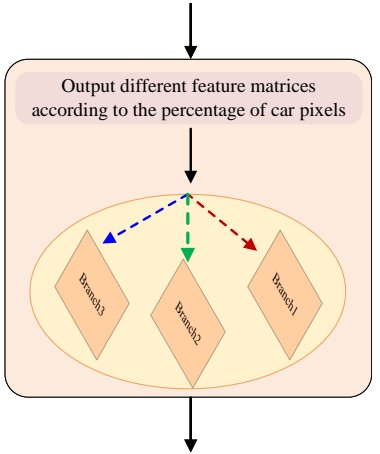

**Figure 3.** The architecture of the dynamic module.

### 2.3. Context Extractor Module

The ASPP module used in the paper, shown in Figure 4, consists of three main parts, $1 \times 1$ convolution, pooling pyramid, and ASPP pooling.

1. The $1 \times 1$ convolution is a linear transformation of the input feature map.
2. The pooling pyramid consists of multiple pooling layers, which are able to reduce the spatial information. In EAD-Net, the dilation rates of the pooling pyramid layers are

set to 6, 12, and 18, respectively. These dilation rates allow the pyramid to capture multi-scale spatial information without resolution loss.

3.  ASPP pooling combines the output of the pooled pyramid and the $1 \times 1$ convolution. It is able to combine the multi-scale spatial information from the pooling pyramid with transformed features from the $1 \times 1$ convolution to generate higher-level feature maps. These feature maps contain richer contextual information, which helps the model detect and recognize objects more accurately.

The ASPP module can capture multi-scale spatial information without reducing the resolution of the feature maps, which enables the model to better understand the context of the objects in the image, leading to improved performance and accuracy.

As shown in Figure 5, the channel attention module [41] consists of several components: an input layer, a squeeze layer, an expand layer, and a gate layer.

1.  Input layer: The input layer receives the output of the previous layer or the input feature maps. It applies a $1 \times 1$ convolution to reduce the number of channels and generate a squeeze feature map.
2.  Squeeze layer: The squeeze layer is responsible for aggregating the information from each channel of the squeeze feature map. It usually applies a global average pooling operation to compress the spatial information and obtain a 1D feature vector representing the channel-wise statistics.
3.  Expand layer: The expand layer takes the 1D feature vector from the squeeze layer as input and expands it back into a 2D feature map with the same number of channels as the input feature maps. This is achieved using a $1 \times 1$ convolution operation.
4.  Gate layer: The gate layer is designed to learn the correlation between channels and assign different weights to each channel. It uses a sigmoid activation function to generate a binary mask that emphasizes the important channels and suppresses the irrelevant ones. The resulting weighted feature maps are then combined with the expanded feature maps from the expanded layer.

During training, the channel attention module learns to assign higher weights to channels that contain valuable information and lower weights to channels with less relevant information, which helps the model focus on the most informative channels and suppress noisy or irrelevant ones, leading to better training and improved performance.

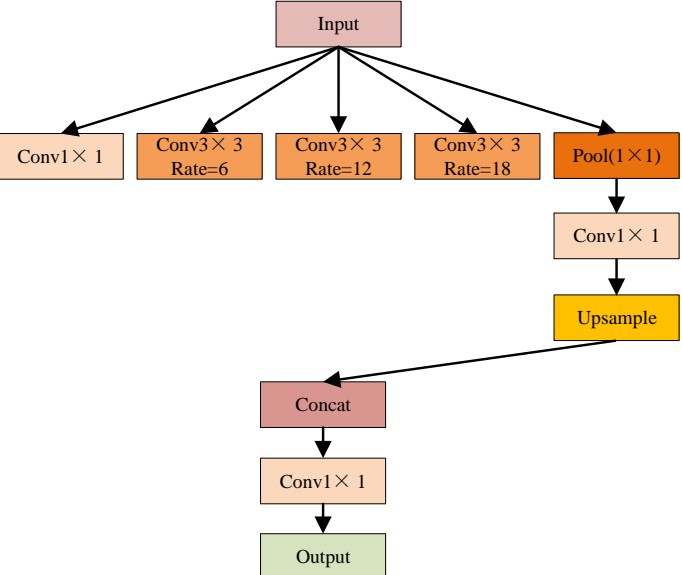

**Figure 4.** The structure of the Adaptive and Selective Perceptual Pooling.

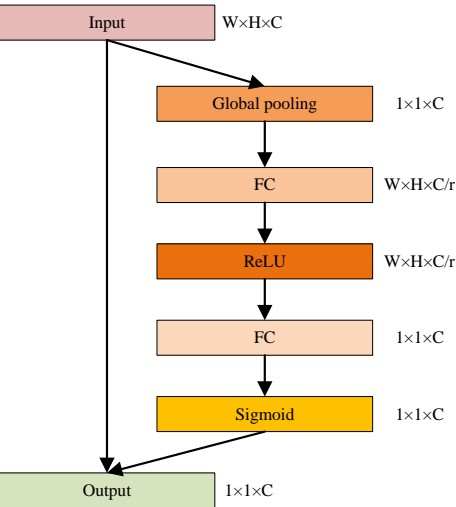

**Figure 5.** The structure of the channel attention module.

*2.4. Decoder Module*

For image semantic segmentation networks, a decoder usually uses one or more up-sampling operations, which may lead to loss of spatial information. To address this issue, an asymmetric decoder is used in EAD-Net that does not use jump connections, but still fuses the low-level features of the encoder efficiently, which aims to complement the high-resolution information by recovering more information on the edges of objects in RSIs.

The proposed decoder's architecture consists of the following steps:

1.  $3 \times 3$ convolutions: To reduce redundant features, $3 \times 3$ convolutions are applied to the fused low-level features from the encoder. This step helps to retain essential spatial information while eliminating unnecessary details.
2.  Fusion with high-level information: The output of the $3 \times 3$ convolutions is then combined with the high-level context information extracted from the context extraction module. This step enables the fusion of both spatial and semantic information, resulting in a more comprehensive representation of the input image.
3.  $1 \times 1$ convolutions: Finally, $1 \times 1$ convolutions are used to reduce the number of channels, leading to a more compact feature representation.

The design of the decoder can effectively reduce the time of fusing features using skip connection and have less computation while supplementing the same amount of spatial information. The decoder can be expressed as follows.

$$out = (F_{low} + F_{high}) \times C^{(1)} \tag{1}$$

$$F_{low} = (F_1 + F_2 + F_3 + F_4) \times C^{(3)} \times C^{(1)} \tag{2}$$

$$F_{high} = Up_4(Up_4(F_5)) \times C^{(1)} \tag{3}$$

where $F_1$, $F_2$, $F_3$, and $F_4$ stand for the output of encoders in the different stages, respectively. $F_5$ represents the output of the semantic extraction module. $C^{(1)}$ and $C^{(3)}$ stand for $1 \times 1$ and $3 \times 3$ convolution, respectively. $Up_4$ represent $4\times$ up-sampling.

*2.5. Loss Function*

Class imbalance is a prevalent issue in classification problems, which can lead to biased learning and poor performance of the model. To address this problem, we used the Cost-sensitive Focal Loss (CFL) [42] function to handle class imbalance in a multi-class pixel-level classification model. The CFL function is a modified version of the original Focal Loss (FL) [43,44] function, which is designed to address the class imbalance problem.

The CFL function is defined as follows:

$$Loss_{CFL} = -\alpha \times \omega_c \times \log(p_i) + FL \tag{4}$$

$$\omega_c = 1 - (r_1 - r_2) \tag{5}$$

$$r_i = rank(p_i), i \in [1, \text{number\_class}] \tag{6}$$

$$FL = -(1 - p_i)^2 \times \log(p_i) \tag{7}$$

where $p_i$ is the softmax probability, and $rank(p_i)$ represents the descending order of the model prediction probability vector. $\alpha$ is the reconciliation parameter, set to 2.5 based on experience.

## 3. Implementation

### 3.1. Dataset Description

The first benchmark dataset used in this paper is the Vaihingen urban classification dataset from ISPRS, which covers a relatively small village with many individual houses and small buildings, contains 3-band IRRG (infrared, red, and green) image data, and corresponding DSM (digital surface model) and NDSM (normalized digital surface model) data. There are a total of 33 images, each approximately 2500 × 2000 pixels in size, with a ground sampling distance (GSD) of around 9 cm. Out of these, 16 images have available surface truth, while the ground truth for the remaining 17 images is unattainable. The dataset includes 6 labeled categories: impervious surface (Imp_surf), building, low vegetation (Low_veg), tree, car, and clutter.

The Potsdam dataset is the second used in this paper, which is a historic city with a large built-up district, narrow streets, and dense settlements. The dataset contains 3-band IRRG (infrared, red, and green) data, RGB (red, green, blue), 4-band RGBIR (red, green, blue, infrared), and corresponding DSM. In total, there are 38 images of 6000 × 6000 pixels with a GSD of ≈5 cm, of which 24 images have the ground truth available and 14 images are unlabeled. In this paper, 14 original images (numbered 1, 3, 11, 13, 15, 17, 21, 23, 26, 28, 30, 32, 34, 37), and 24 original images (numbered 2_10, 2_11, 2_12, 3_10, 3_11, 3_12, 4_10, 4_11, 4_12, 5_10, 5_11, 5_12, 6_7, 6_8, 6_9, 6_10, 6_11, 6_12, 7_7, 7_8, 7_9, 7_10, 7_11, 7_12) in Vaihingen and Potsdam, respectively, are selected as the training and validation sets and the remainder is the test data. An original patch and its corresponding ground truth in the ISPRS Vaihingen and Potsdam datasets, respectively, are shown in Figure 6.

As shown in Table 1, we can see that the two datasets have an imbalanced distribution of categories, in which the percentage of car and clutter is particularly small. In the subsequent experiments, some measures have been taken to suppress the unbalanced distribution.

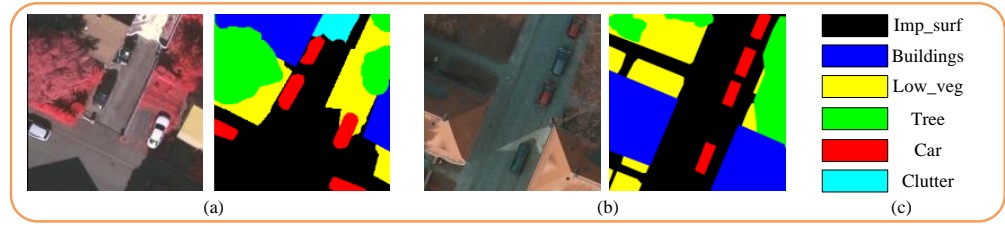

**Figure 6.** A sample of a patch from the ISPRS Vaihingen dataset and its corresponding ground truth in (**a**), a sample of a patch from the ISPRS Potsdam dataset and its corresponding ground truth in (**b**), and the different categories labeled by different colors in (**c**).

**Table 1.** The percentage of each category in the Vaihingen and Potsdam datasets.

| Datasets | Impervious Surface | Building | Low Vegetation | Tree | Car | Clutter |
|---|---|---|---|---|---|---|
| Vaihingen | 28 | 29 | 21 | 23 | 1.2 | 0.8 |
| Potsdam | 33 | 26 | 22 | 14 | 1.3 | 3.7 |

### 3.2. Data Preprocessing

In this paper, we address the issue of large image sizes and insufficient datasets by cropping the original into smaller patches and performing various data augmentation techniques. These steps aim to improve the model's performance while reducing the risk of overfitting. The original images and corresponding labels of the training and validation sets are finally cropped into patches of size $256 \times 256$ using sliding windows (with a slide step of 128) in the Vaihingen dataset, and $512 \times 512$ (with a slide step of 256) in the Potsdam dataset, following the overlapping approach described in this paper. Then these patches are all randomly rotated, with added noise and other data augmentation to enlarge the number of the training dataset to train an effective model.

By implementing these strategies, we address the challenges of large image sizes and insufficient datasets. The cropping and data augmentation techniques enable training an effective deep neural network while reducing the risk of overfitting. This approach can lead to better performance on the target task and a more robust model that can handle variations in the input data.

### 3.3. Implementation Details

In this section, we provide a detailed description of the experimental setup used in our study. We employed a server equipped with 8 GTX 1080Ti graphics cards to handle the computational demands of the network training. To ensure stability and parallel processing without abnormalities under multiple threads, we confined the network training to a single GPU.

The software environment included Ubuntu 16.04, Python 3.8, PyTorch 1.10.0, CUDA 11.2, and a GPU driver version 460.80. The network training batch size was set to 8, which facilitated efficient training and minimized the overhead of data movement.

The initial learning rate was set to 0.01, and it was adjusted accordingly during the training process based on the training situation. Eventually, the learning rate was fixed to 0.0001. We employed a polynomial decay method for adjusting the learning rate. The network optimization method utilized momentum with a value of 0.9, which helped accelerate the convergence of the algorithm. Weight decay was initially set to 0.001 and later fixed to 0.0005 to reduce the magnitude of weights and improve the model's generalization capabilities.

During each iteration, the network was trained for 100 epochs using the CFL optimizer (details provided in Section 2.5) and a polynomial learning strategy. To guarantee the accuracy and reliability of the experimental results, we did not utilize pre-trained weights from other datasets on the feature extraction network for transfer learning. Instead, we selected the model weights with the best training effect for the next iteration after each iteration.

### 3.4. Evaluation Metrics

To assess the quantitative performance, two overall benchmark metrics are used, i.e., mIoU, and mPA, which can comprehensively compare the performance of different methods. The calculation formulas are shown as follows.

$$IoU = \sum_{i=0}^{k} \frac{TP}{FN + FP + TP} \tag{8}$$

$$PA = \sum_{i=0}^{k} \frac{TP + TN}{TP + FN + FP + TN} \tag{9}$$

$$mIoU = \frac{1}{k+1} \Sigma_{i=0}^{k} \frac{TP}{FN + FP + TP} \tag{10}$$

$$mPA = \frac{1}{k+1} \sum_{i=0}^{k} \frac{TP + TN}{TP + FN + FP + TN} \tag{11}$$

where $k$ is the number of categories, $TP$, $FN$, $FP$ and $TN$ are the number of true positive, false negative, false positive, and true negative, respectively.

## 4. Experimental Comparison and Analysis

### 4.1. Comparison with the State-of-the-Art

In our experiments, we evaluate seven state-of-the-art semantic labeling methods for comparison with our proposed approach. A summary of these comparison methods is provided below:

1.  FCN [21]: FCN (Fully Convolutional Network) was the first method to apply deep convolutional neural networks (DCNNs) for image classification to the semantic segmentation task.
2.  DeepLab series [26,27]: This series of models has been highly successful and widely adopted for image semantic segmentation.
3.  LR-ASPP [31]: A lightweight network suitable for mobile deployment applications.
4.  SegNet [24]: A classical semantic segmentation network featuring an encoder-decoder architecture.
5.  U-Net [22]: Another encoder-decoder structure commonly used for medical image segmentation.
6.  PSPNet [25]: Pyramid Scene Parsing Network, which is based on the fully convolutional idea of FCN and represents one of the successful improved segmentation models built upon FCN.

By comparing our proposed method with these state-of-the-art approaches, we aim to demonstrate the effectiveness and competitiveness of our solution in the field of semantic segmentation.

### 4.2. Experimental Results on the Vaihingen Dataset

4.2.1. Quantitative Evaluation

After refining the above details and establishing consistent training parameters, the ultimate overall training outcomes for the presented models following uneven time training are exhibited in Table 2.

The quantitative results of the mPA and mIoU of all methods with different backbones are shown in Table 2. Table 2 shows that U-Net achieves the lowest mPA and mIoU of 64.84% and 61.54%, respectively, and indicates that it is ill-suited for the remote sensing benchmark dataset due to its large size, multiple semantic categories, and sensitivity to low-level features. The second lowest is PSP-Net, with 67.96% and 70.15%, respectively. DeepLabV3+, similar to Lite R-ASPP in using MobileNetV3 as the backbone, also achieves good segmentation outcomes. These results indicate that on the ISPRS dataset, EAD-Net achieves the best performance in semantic segmentation. The mPA and mIoU of EAD-Net are 92.27% and 87.38%, with 8% higher mIoU than DeepLabV3. However, both EAD-Net and DeepLabV3 use ResNet50 as the backbone network. It can be inferred that the dynamic path selection and channel attention mechanisms introduced in EAD-Net are feasible, and the combination of them can improve the performance of segmentation accuracy. The validation thereof will be shown in the ablation experimental results.

The quantitative results of PA for each category and mPA are shown in Table 3, and the IoU for each category and mIoU are shown in Table 4. As can be seen in Tables 3 and 4, the evaluation metrics PA and IoU are generally positively correlated. Compared to other SOTA methods, EAD-Net achieves the best results in terms of segmentation accuracy of most categories. Buildings and impervious surfaces with larger scales and regular shapes are less difficult to segment. FCN-8s, DeepLabV3, SegNet, and DeepLabV3+ all achieve segmentation accuracies (mIoU) of more than 85%. It shows that all these methods can correctly recognize the features of these two categories and achieve pixel-level classification. Cars are difficult to segment due to their small scale and ease of occlusion, and the segmentation accuracies of all the methods are not high, while EAD-Net has achieved the highest accuracy of 84.65 on mIoU. It can therefore be deduced that our method has

better performance in feature extraction for small-scale targets in RSIs, resulting in higher segmentation accuracies. Trees and low vegetation, both of which are similar in color and geographically close to each other, make segmentation more difficult. The experimental results in Tables 3 and 4 show that the segmentation accuracies of these two categories do not differ much.

**Table 2.** Quantitative comparison (%) with deep learning models on ISPRS Vaihingen dataset, where the values in bold are the best. mPA: mean pixel accuracy. mIoU: mean intersection over union.

| | Methods | | | | | | | |
|---|---|---|---|---|---|---|---|---|
| | **FCN-8s [21]** | **DeepLabV3 [26]** | **LR-ASPP [31]** | **SegNet [24]** | **U-Net [22]** | **PSP-Net [25]** | **DeepLabV3+ [27]** | **EAD-Net** |
| Backbone | ResNet50 | ResNet50 | MobileNetV3 | ResNet50 | MobileNetV3 | ResNet50 | MobileNetV3 | ResNet50 |
| mPA | 91.93 | 92.01 | 78.90 | 82.14 | 64.84 | 67.96 | 85.87 | **92.27** |
| mIoU | 86.08 | 86.11 | 69.98 | 74.86 | 61.54 | 70.15 | 76.61 | **87.38** |

**Table 3.** Quantitative evaluation results on ISPRS Vaihingen dataset. where the values in bold are the best. PA: pixel accuracy. mPA: mean pixel accuracy.

| Method / Category | PA(%) | | | | | | | |
|---|---|---|---|---|---|---|---|---|
| | **FCN-8s [21]** | **DeepLabV3 [26]** | **LR-ASPP [31]** | **SegNet [24]** | **U-Net [22]** | **PSP-Net [25]** | **DeepLabV3+ [27]** | **EAD-Net** |
| Imp_ surf | 94.26 | 94.03 | 87.88 | 91.73 | 75.11 | 70.28 | 90.79 | 94.28 |
| Building | 94.83 | 95.11 | 91.17 | 85.92 | 76.14 | 77.63 | 95.34 | 95.66 |
| Low_ veg | 89.77 | 89.68 | 81.22 | 82.22 | 66.14 | 73.22 | 84.38 | 89.87 |
| Tree | 90.76 | 91.27 | 82.86 | 87.83 | 68.07 | 63.29 | 84.93 | 91.43 |
| Car | 85.79 | 85.04 | 61.66 | 54.55 | 50.32 | 59.47 | 69.47 | 88.91 |
| Clutter | 96.18 | 96.94 | 68. 61 | 90.59 | 53.23 | 63.88 | 90.29 | 96.17 |
| **mPA (%)** | 91.93 | 92.01 | 78.90 | 82.14 | 64.84 | 67.96 | 85.87 | **92.72** |

**Table 4.** Quantitative evaluation results on ISPRS Vaihingen dataset, where the values in bold are the best. IoU: intersection over union. mIoU: mean intersection over union.

| Method / Category | IoU(%) | | | | | | | |
|---|---|---|---|---|---|---|---|---|
| | **FCN-8s [21]** | **DeepLabV3 [26]** | **LR-ASPP [31]** | **SegNet [24]** | **U-Net [22]** | **PSP-Net [25]** | **DeepLabV3+ [27]** | **EAD-Net** |
| Imp_ surf | 88.35 | 88.84 | 78.39 | 86.44 | 69.24 | 70.16 | 84.77 | 86.70 |
| Building | 91.33 | 91.55 | 83.64 | 85.07 | 64.56 | 78.68 | 90.63 | 91.64 |
| Low_ veg | 82.31 | 82.52 | 69.10 | 72.72 | 61.37 | 69.74 | 72.90 | 83.45 |
| Tree | 81.66 | 82.37 | 67.82 | 72.93 | 57.26 | 68.52 | 70.66 | 83.12 |
| Car | 78.04 | 77.36 | 53.07 | 50.55 | 46.11 | 60.39 | 59.52 | 84.65 |
| Clutter | 94.79 | 94.04 | 67.88 | 81.42 | 70.70 | 73.41 | 81.17 | 94.69 |
| **mIoU (%)** | 86.08 | 86.11 | 69.98 | 74.86 | 61.54 | 70.15 | 76.61 | **87.38** |

### 4.2.2. Qualitative Evaluation

Given the poor segmentation results of the U-Net and PSPNet networks and their inapplicability to the dataset, they are not discussed in subsequent sections. Figures 7–9 present the comparison of segmentation effects in three scenarios using randomly selected patches from the test and validation datasets.

As illustrated in Figure 7, the majority of pixels in these patches are occupied by buildings and impervious surfaces. It is evident that different approaches have varying impacts on the boundaries. All the aforementioned methods perform effectively for dividing buildings and impervious surfaces, given that most of them have regular shapes and distinct spectral characteristics. In Figure 7, by comparing the rectangular labeled areas in each row, we can observe that EAD-Net demonstrates a superior ability to delineate the edges and corners of buildings and impervious surfaces. This results in more precise and smoother boundary segmentation, bringing the outcomes closer to the ground truth, which shows that EAD-Net not only improves the segmentation accuracy of small-scale targets but also the visual effect after segmentation for buildings and impervious surfaces with

regular shapes. That which we demonstrate in Figure 7 is consistent with the quantitive results in Tables 2–4.

As depicted in Figure 8, a significant portion of the pixels in these patches are occupied by trees and low vegetation. Most of the methods examined perform well in segmenting large-scale trees or extensive low vegetation. However, for small-scale trees, the segmentation accuracy is less satisfactory, and the contours are not as precise. This is attributed to the fact that trees and low vegetation have similar colors and geographical locations, making their spectral characteristics challenging to differentiate. Furthermore, some trees are planted on top of low vegetation, which exacerbates the segmentation task. By examining the rectangle labeled area in Figure 8, it can be seen that EAD-Net yields the best segmentation performance for both large-scale trees and low vegetation. This is likely due to its channel attention mechanism to handle the similarities in color and location between trees and low vegetation, as well as its effectiveness in differentiating small-scale trees from their surrounding environments.

As illustrated in Figure 9, a significant portion of the pixels in these patches are occupied by cars. When comparing the performance of various methods, EAD-Net stands out as it successfully segments nearly all small-scale objects in the image, producing smooth and precise segmentation edges that are remarkably close to the ground truth. This outstanding performance can be attributed to the dynamic module designed specifically to enhance the segmentation accuracy of small-scale objects in EAD-Net. The module effectively addresses the difficulties encountered in segmenting small objects, such as cars, which is evident from the visual results. The improved segmentation accuracy of small-scale objects in EAD-Net demonstrates its superiority over other methods and establishes it as a promising candidate for applications requiring accurate segmentation of dense, small-scale objects.

By analyzing the quantitative and qualitative results, it is evident that the visual effectiveness of segmentation is directly related to the Mean Intersection over Union (MIoU) score. EAD-Net outperforms other methods in terms of segmentation performance, showcasing the effectiveness of the proposed method.

In this paper, the finest-tuned model weights are preserved, and the original image is resized to $256 \times 256$ pixels before prediction. Upon completion of the prediction, the patches are reconstructed to their original image, as shown in Figure 10.

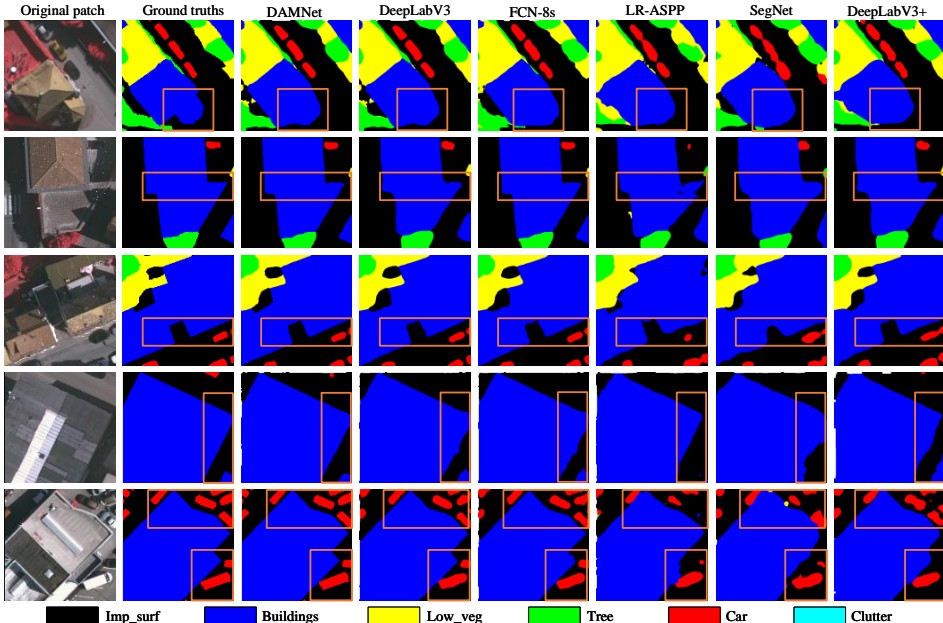

**Figure 7.** Visual comparisons with deep learning models of local evaluation on ISPRS Vaihingen dataset.

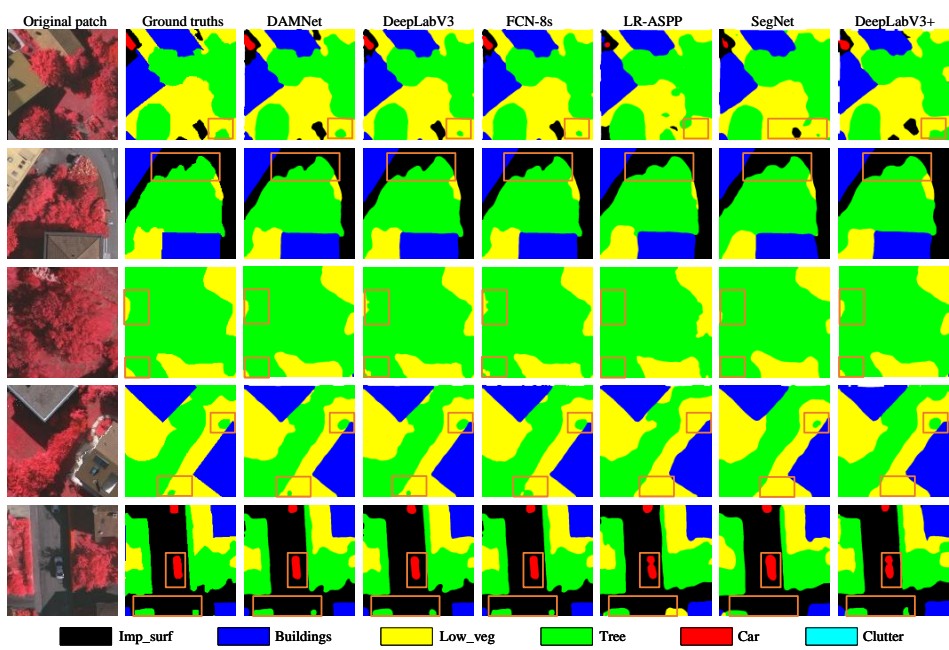

**Figure 8.** Visual comparisons with deep learning models of local evaluation on ISPRS Vaihingen dataset.

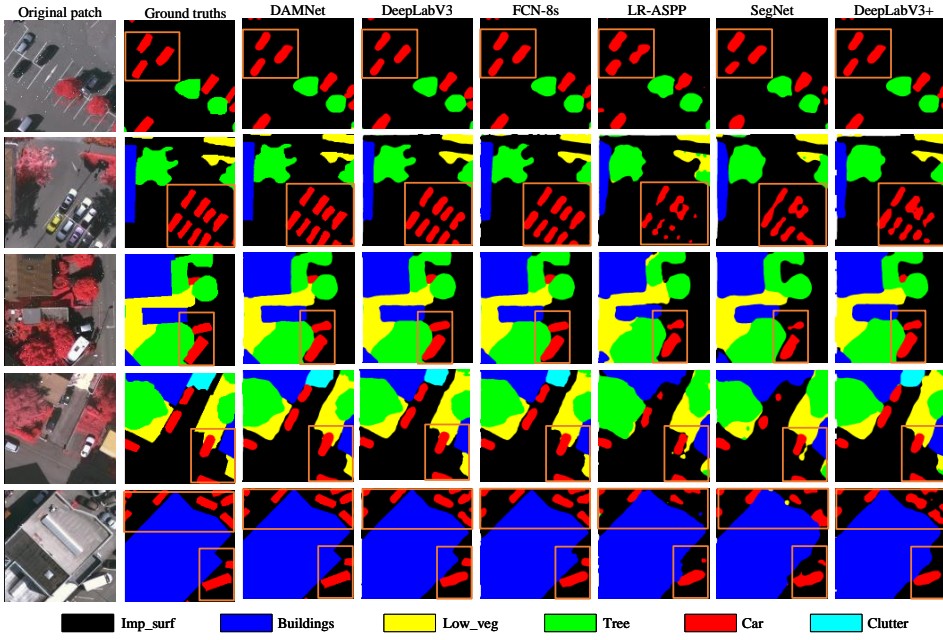

**Figure 9.** Visual comparisons with deep learning models of local evaluation on ISPRS Vaihingen dataset.

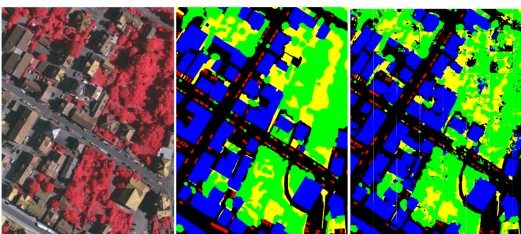

**Figure 10.** Classification maps of an original image in the Vaihingen dataset.

*4.3. Experimental Results on the Potsdam Dataset*

4.3.1. Quantitative Evaluation

About experimental results on the Potsdam dataset, the experimental setup and parameters remain consistent with 4.2. However, due to the presence of larger buildings and wide streets in the Potsdam dataset, the original images were cropped to $512 \times 512$ pixels. The ultimate overall training outcomes achieved by the aforementioned models after uneven time training are exhibited in Table 5.

As depicted in Table 5, EAD-Net emerges as the top-performing model, achieving respective mPA and mIoU scores of 96.27% and 93.10%, respectively. DeepLabV3+, utilizing MobileNetV3 as the backbone, ranks second only to EAD-Net among the compared networks. FCN-8s and DeepLabV3 also deliver outstanding segmentation results, surpassing other networks in terms of mPA and mIoU metrics.

Tables 6 and 7 present the performance results of various models on the ISPRS Potsdam dataset. It is evident that the mPA and mIoU scores stand at 96.27% and 93.10%, respectively, and have achieved SOTA performance. Except for the EAD-Net model, DeepLabV3+ has the best performance, followed by FCN and LR-ASPP.

**Table 5.** Quantitative comparison (%) with the state-of-the-art deep models on the Potsdam dataset, where the values in bold are the best. mPA: mean pixel accuracy. mIoU: mean interSection over union.

| | **Methods** | | | | | | | |
|---|---|---|---|---|---|---|---|---|
| | **FCN-8s [21]** | **DeepLabV3 [26]** | **LR-ASPP [31]** | **SegNet [24]** | **U-Net [22]** | **PSP-Net [25]** | **DeepLabV3+ [27]** | **EAD-Net** |
| Backbone | ResNet50 | ResNet50 | MobileNetV3 | ResNet50 | MobileNetV3 | ResNet50 | MobileNetV3 | ResNet50 |
| mPA | 89.29 | 88.29 | 84.75 | 87.85 | 66.82 | 90.37 | 94.69 | **96.27** |
| mIoU | 81.80 | 79.92 | 75.52 | 80.65 | 60.35 | 65.58 | 89.93 | **93.10** |

**Table 6.** Quantitative comparison (%) of PA with the state-of-the-art deep models on the ISPRS Potsdam dataset, where the values in bold are the best.

| Method<br>Category | **PA(%)** | | | | | | | |
|---|---|---|---|---|---|---|---|---|
| | **FCN-8s [21]** | **DeepLabV3 [26]** | **LR-ASPP [31]** | **SegNet [24]** | **U-Net [22]** | **PSP-Net [25]** | **DeepLabV3+ [27]** | **EAD-Net** |
| Imp_ surface | 93.32 | 92.49 | 90.67 | 93.49 | 76.60 | 94.51 | 96.89 | 97.82 |
| Building | 97.28 | 97.10 | 96.21 | 97.64 | 77.23 | 97.99 | 98.67 | 99.15 |
| Low_ veg | 89.54 | 88.81 | 87.91 | 89.61 | 65.14 | 90.90 | 94.96 | 96.63 |
| Tree | 85.38 | 83.56 | 76.31 | 83.19 | 62.39 | 87.60 | 92.52 | 94.86 |
| Car | 90.24 | 87.71 | 85.09 | 88.72 | 58.77 | 87.26 | 92.03 | 93.67 |
| Clutter | 79.98 | 80.05 | 72.31 | 74.42 | 60.78 | 83.95 | 92.92 | 95.50 |
| **mPA (%)** | 89.29 | 88.29 | 84.75 | 87.85 | 66.82 | 90.37 | 94.69 | **96.27** |

**Table 7.** Quantitative comparison (%) of IoU with the state-of-the-art deep models on the ISPRS Potsdam dataset, where the values in bold are the best.

| Method<br>Category | **IoU(%)** | | | | | | | |
|---|---|---|---|---|---|---|---|---|
| | **FCN-8s [21]** | **DeepLabV3 [26]** | **LR-ASPP [31]** | **SegNet [24]** | **U-Net [22]** | **PSP-Net [25]** | **DeepLabV3+ [27]** | **EAD-Net** |
| Imp_ surf | 87.50 | 86.73 | 83.44 | 87.17 | 63.49 | 89.25 | 93.61 | 95.51 |
| Building | 94.49 | 94.15 | 91.87 | 94.58 | 70.22 | 95.59 | 97.55 | 98.28 |
| Low_ veg | 79.34 | 77.70 | 73.27 | 78.43 | 74.04 | 82.40 | 89.93 | 93.21 |
| Tree | 75.56 | 72.96 | 66.25 | 73.76 | 65.38 | 79.04 | 86.40 | 90.88 |
| Car | 82.95 | 78.47 | 75.10 | 82.14 | 58.63 | 79.19 | 86.93 | 88.96 |
| Clutter | 70.96 | 69.80 | 63.02 | 67.05 | 55.19 | 75.91 | 87.49 | 91.73 |
| **mIoU (%)** | 81.80 | 79.97 | 75.49 | 80.52 | 64.49 | 83.56 | 90.35 | **93.10** |

4.3.2. Qualitative Evaluation

In Figure 11, a large area of pixels in the patches is taken up by buildings (black) and impervious surfaces (blue), which are structurally defined and larger, making them relatively easy to segment. Most models accurately segment these categories. EAD-Net

posts superior visual results with precise segmentation, clear boundaries, and smooth results. The segmentation outcomes vary with different architectural modules of the same backbone network, aligning with previous quantitative findings.

In Figure 12, many pixels in these patches are occupied by trees and low vegetation, represented by green and yellow colors, respectively. Due to their similar colors and geographical locations, trees often stand within low vegetation, making accurate segmentation of both challenging. By comparing the second and third columns in Figure 12, it can be seen that our model performs remarkably well in segmenting the contours of trees and low vegetation, particularly those trees situated within low vegetation, showcasing clear boundaries.

In Figure 13, the objects marked in red are cars. Given their small appearance and dense distribution, cars are prone to being obscured and misclassified in remote sensing images, making them hard to segment accurately. When comparing the second and third columns in Figure 13, it becomes apparent that our segmentation method exhibits virtually no difference in visual effectiveness compared to the ground truth.

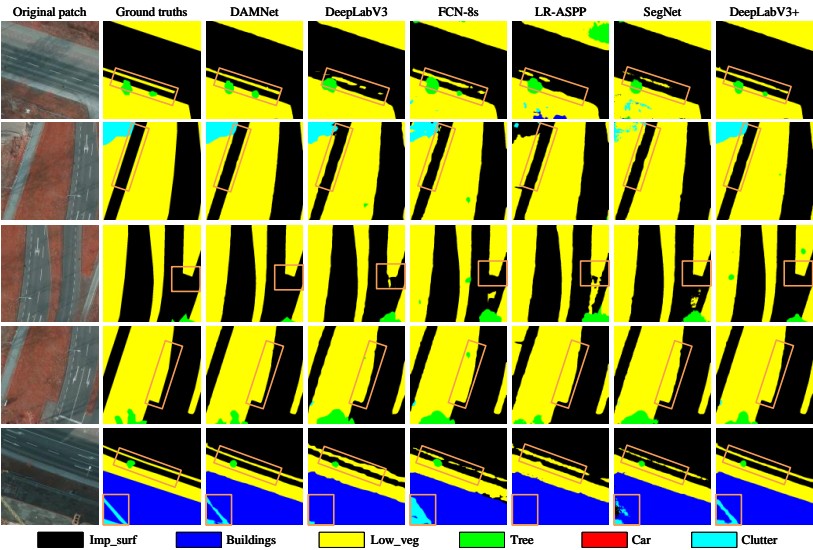

**Figure 11.** Visual comparisons with deep learning models of local evaluation on ISPRS Potsdam dataset.

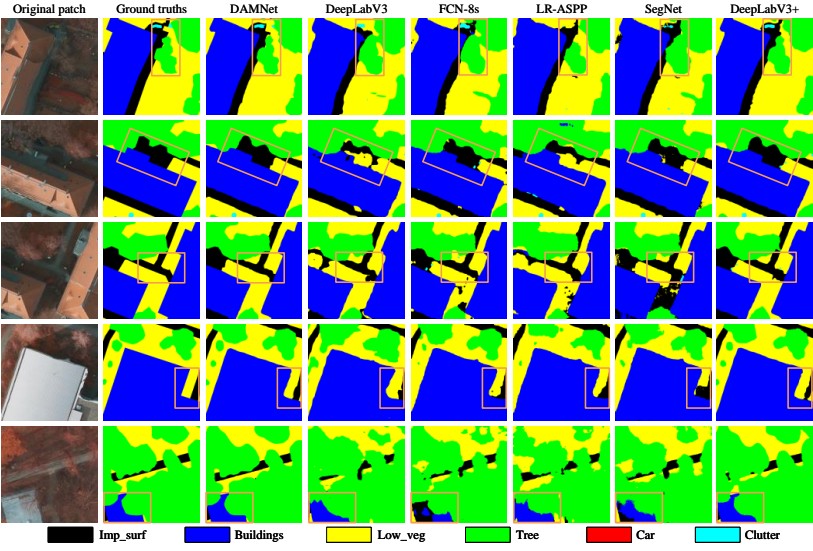

**Figure 12.** Visual comparisons with deep learning models of local evaluation on ISPRS Potsdam dataset.

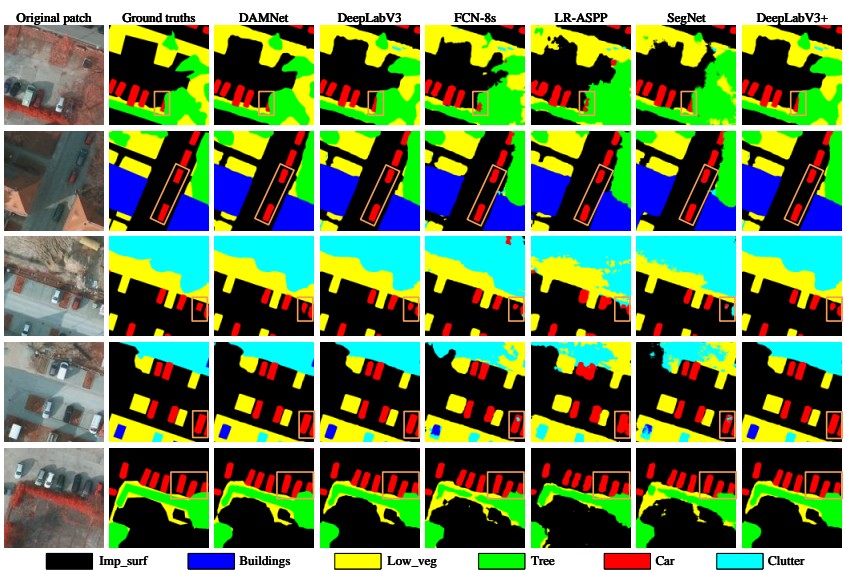

**Figure 13.** Visual comparisons with deep learning models of local evaluation on ISPRS Potsdam dataset.

### 4.4. Ablation Study for EAD-Net

According to the presentation and discussion of the experimental results in Section 4.3, it can be seen that EAD-Net achieved the best segmentation results, which depends on the integration of its dynamic routing selection module and the channel attention mechanism, which makes it possible to significantly improve the segmentation accuracy of the small-scale targets without degrading the segmentation accuracy of the other categories, so that it achieves the SOTA performance. In order to verify the findings in Section 4.3, we implemented extensive ablation experiments.

#### 4.4.1. Ablation Study for Parameters

To balance the consideration of computation and accuracy of the model, EAD-Net used ResNet50 as the backbone network. The results of the ablation experiments are shown in Table 8, which shows the use of four different backbone networks, and corresponding the number of floating point operations per second (FLOPs) and the number of parameters (Params).

As can be seen from Table 8, with ResNet101 as the backbone, the accuracy is indeed higher than that with ResNet50, but only by just over 1%. However, the number of FLOPs and parameters are indeed more than 1.5 times that of the latter, respectively. Weighting the amount of computation and parameters, the segmentation accuracy of ResNet50 as the backbone network is ideal in our experiments.

**Table 8.** Ablation study of ISPRS Vaihingen and Potsdam datasets about four different backbone networks.

|  |  | Backbone | | | |
| --- | --- | --- | --- | --- | --- |
|  |  | **ResNet 18** | **ResNet 34** | **ResNet 50** | **ResNet 101** |
| Vaihingen | FLOPs(G) | 31.854047232 | 41.543938048 | 46.984892416 | 66.486308864 |
|  | Params(M) | 16.803289 | 26.911449 | 40.940505 | 59.932633 |
|  | mIoU(%) | 84.21 | 85.66 | 87.38 | 88.93 |
| Potsdam | FLOPs(G) | 36.45278166 | 43.18914782 | 48.33298745 | 70.83546797 |
|  | Params(M) | 18.065447 | 19.136658 | 42.588741 | 64.2938174 |
|  | mIoU(%) | 88.29 | 90.06 | 93.10 | 94.37 |

4.4.2. Ablation Study of Each Module

To validate the effectiveness of the SE module and Dynamic routing module in EAD-Net, we conducted corresponding ablation experiments on the ISPRS Vaihingen and Potsdam datasets, respectively. The PA and IoU are presented in Table 9 and Table 10, respectively. A "✓" symbol indicates the inclusion of the respective module in EAD-Net.

To validate the effectiveness of the SE and Dynamic routing modules in EAD-Net, we conducted the corresponding ablation experiments on the Vaihingen and Potsdam datasets, respectively. The PA and IoU values are presented in Tables 9 and 10. A "✓" symbol indicates the inclusion of the respective module in EAD-Net.

The analysis of the above experimental results shown in Tables 9 and 10, indicates that upon incorporating ASPP, either removing the Dynamic module and adding the SE module or removing the SE module and adding the Dynamic module has a minimal impact on PA improvement. In EAD-Net, the integration of both modules results in a substantial increase in PA, particularly for small-scale objects, of nearly 3%. This enables the most challenging targets to be segmented with a PA of nearly 90%. This demonstrates that EAD-Net can effectively enhance the segmentation accuracy of HRRSIs, particularly for small-scale objects, which are the most difficult to segment. This is fully consistent with the qualitative experimental results. The experimental results confirm the effectiveness of our method (EAD-Net), which is consistent with the previous experimental results.

**Table 9.** Ablation study of ISPRS Vaihingen dataset

| ASPP | SE Module | Dynamic Module | Category | | | | | | | | | | | |
| | | | Imp_ surf | | Building | | Low_ veg | | Tree | | Car | | Clutter | |
| | | | PA | IoU | PA | IoU | PA | IoU | PA | IoU | PA | IoU | PA | IoU |
| ✓ | | | 90.58 | 83.12 | 91.04 | 88.22 | 85.34 | 78.26 | 85.44 | 78.64 | 80.22 | 81.58 | 92.78 | 90.71 |
| ✓ | ✓ | | 93.27 | 85.66 | 93.51 | 90.17 | 88.06 | 81.48 | 89.14 | 81.18 | 85.96 | 83.44 | 94.49 | 93.06 |
| ✓ | | ✓ | 93.22 | 85.16 | 93.24 | 90.88 | 88.24 | 81.39 | 88.67 | 81.09 | 86.33 | 83.16 | 95.23 | 92.88 |
| ✓ | ✓ | ✓ | **94.28** | **86.70** | **95.66** | **91.64** | **89.87** | **83.45** | **91.43** | **83.12** | **88.91** | **84.65** | **96.17** | **94.69** |

**Table 10.** Ablation study of ISPRS Potsdam dataset.

| ASPP | SE Module | Dynamic Module | Category | | | | | | | | | | | |
| | | | Imp_ surf | | Building | | Low_ veg | | Tree | | Car | | Clutter | |
| | | | PA | IoU | PA | IoU | PA | IoU | PA | IoU | PA | IoU | PA | IoU |
| ✓ | | | 92.33 | 95.08 | 93.74 | 88.22 | 89.54 | 92.68 | 86.33 | 90.41 | 84.72 | 88.33 | 87.42 | 90.71 |
| ✓ | ✓ | | 94.51 | 95.63 | 96.55 | 90.17 | 91.20 | 94.39 | 88.47 | 92.88 | 86.29 | 90.15 | 89.54 | 92.05 |
| ✓ | | ✓ | 94.26 | 96.61 | 96.93 | 90.88 | 90.77 | 94.78 | 89.15 | 93. 60 | 87.11 | 90.56 | 90.26 | 93.66 |
| ✓ | ✓ | ✓ | **97.82** | **95.51** | **99.15** | **98.28** | **96.63** | **93.21** | **94.86** | **90.88** | **93.67** | **88.96** | **95.50** | **91.73** |

4.4.3. Ablation Study of Loss Function

To verify that the CFL function can alleviate the category imbalance problem in RSIs, we also implemented the ablation implementation on ISPRS Vaihingen and Potsdam datasets. The experimental results are displayed in Figure 14 and Figure 15, respectively.

As illustrated in the experimental results in Figures 14 and 15, employing the CFL function demonstrates a slight, yet significant improvement in segmentation performance for each category in the dataset. The enhancement is approximately 0.2%, but for categories with a relatively low pixel percentage, such as cars and clutter, the improvement reaches 0.8%. This suggests that the CFL function is more effective in suppressing the imbalanced distribution of categories compared to the FL function.

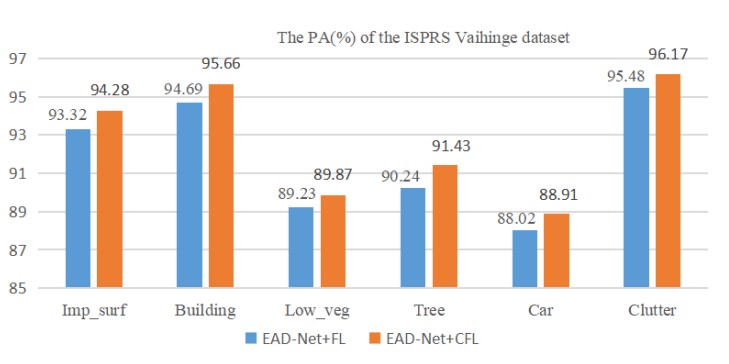

**Figure 14.** Ablation study of ISPRS Vaihingen dataset.

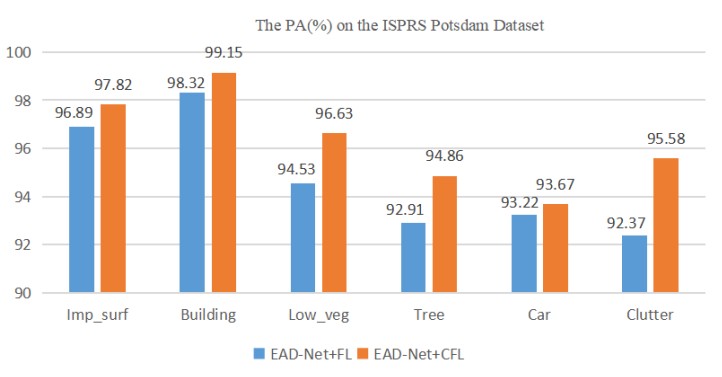

**Figure 15.** Ablation study of ISPRS Potsdam dataset.

## 5. Conclusions

EAD-Net proposes a dynamic routing module that integrates a channel attention mechanism that can be used as a feature enhancement module, which dynamically adjusts the network routing based on the number of pixels of small-scale objects, such as cars, thus improving the accuracy. The advantage has been visually confirmed. To improve the segmentation accuracy of different scale objects in RSIs, ASPP is added to EAD-Net as a context extractor to obtain multi-scale context information without increasing the number of parameters. In addition, an asymmetric decoder is used to reduce redundant information while preserving spatial information without reducing the sampling rate. Extensive experimental results conducted on the ISPRS Vaihingen and Potsdam 2D semantically labeled datasets show that EAD-Net effectively improves the segmentation accuracy in various categories, especially in small-scale objects by nearly 3%. Thus, EAD-Net achieves SOTA performance.

In the next work, first, we could use an effective dynamic routing mechanism for all the categories in the dataset to improve the accuracy of all the categories by dynamically choosing the appropriate paths according to the number of pixels according to the number of pixels belonging to different categories. Secondly, we could also use ResNet101 as the backbone network to implement our experiment to improve the segmentation accuracy of the model. The above works will be completed when the experimental conditions, such, as the graphics card, CPU, GPU, etc., allow it and validate the performance of our model on more datasets.

Analyzing from the perspective of reducing the afford of labeled data, we can try to use semi-supervised learning methods, or unsupervised learning methods to improve the segmentation accuracy.

**Author Contributions:** Writing—original draft, Q.H.; Writing—review and editing, F.W. and Y.L. All authors have read and agreed to the published version of the manuscript.

**Funding:** This research was funded by the National Natural Science Foundation of China (No. 62271400) and the Shaanxi Provincial Key R&D Program, China (No. 2023-GHZD-02).

**Data Availability Statement:** Data available in a publicly accessible repository, The original data presented in the study are openly available in ISPRS at https://www.isprs.org/education/benchmarks/UrbanSemLab/semantic-labeling.aspx.

**Conflicts of Interest:** The authors declare no conflicts of interest.

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
