# Peer review of "EAD-Net: Efficiently Asymmetric Network for Semantic Labeling of High-Resolution Remote Sensing Images with Dynamic Routing Mechanism"

_remotesensing, doi:10.3390/rs16091478_

Round 1
Reviewer 1 Report
Comments and Suggestions for Authors
Here are several suggestions:
1. Including a brief summary in the abstract that quantifies the performance and complexity improvements offered by the proposed algorithm would be valuable for readers. Additionally, please capitalize the word 'concurrently' at the beginning of the sentence in the abstract for grammatical correctness.
2. In the introduction's second paragraph, where conventional methods are discussed, it would be beneficial to acknowledge the existence of efficient unsupervised methods, such as doi.org/10.48550/arXiv.2312.15447 and some others, which achieve quick and accurate results without requiring labels. Their main limitation should be an excessive number of required parameters. Citing or reviewing these methods could provide a fairer and more comprehensive overview of the current landscape. Also, other references on deep networks designed for HRRSIs can be included in lines 41–61 to enrich the discussion as well.
3. Using an enumerate environment in LaTeX for lists would give a better visualization, lines 71–79, 155–161, 203–217, 352–364.
4. Could you clarify the use of the asterisk "∗" in Equation (3) as opposed to the multiplication symbol "\times" used in Equations (1-2)? Ensuring consistency in notation will aid in reader comprehension. Check the other equations in the paper as well.
5. For a comprehensive assessment of model complexity, consider calculating FLOPs across all methods. Additionally, comparing results across networks with a consistent backbone, such as using ResNet50 or MobileNetV3 for all (just like DeepLabV3 and DeepLabV3+), would lend more persuasive power to the comparative analysis if time permits. This suggestion is contingent on the availability of relevant code and time resources, aiming to satisfy my curiosity rather than being a strict requirement.
6. Conducting ablation studies on different backbones, such as ResNet18, ResNet32, ResNet50, and ResNet101, with your method is recommended to validate the efficiency and effectiveness of ResNet50 as the optimal choice. This comparative analysis would strengthen the evidence for ResNet50's selection.
Overall, the manuscript is well-composed and presents its findings clearly. However, to further improve its contribution to the field, I recommend broadening the introduction with a more comprehensive review of related works and incorporating additional experiments. These enhancements would deepen the paper's context and substantiate its claims more robustly.
Comments on the Quality of English LanguageMinor edits needed.
Reviewer 2 Report
Comments and Suggestions for Authors
1. Has the model design in this paper referenced other works? Where can the most significant innovation of this paper be reflected, and what are the bases for the innovation?
2. Does this article address certain practical issues from the perspective of remote sensing research, rather than simply obtaining satisfactory results solely based on open-source datasets?
3. What remote sensing data sources were used in this article? And did it compare the effects of different remote sensing data?
4. There is already a segmentation network called EAD-Net for medical image segmentation, is there a connection between this model and your work?
5. Section 2.2 describes the important role that the Feature Augmentation structure plays in the model, but does not explain how it works clearly enough.
6. In section 3.2, the dataset images are cropped to 256x256 and 512x512, please elaborate on the image cropping methodology, a lot of similar work has been done with larger size images in the same datasets, can you clarify if there is a link between smaller image sizes and model performance?
7. There is some correlation between the difference in semantic segmentation performance and the difference in the number of model parameters listed in Table2, do you think it is because the larger model leads to better results?
Comments on the Quality of English Language1. Although this paper is about semantic segmentation of remotely sensed images, the actual content has no real relevance to remote sensing research and is more like a deep learning study.
2. Especially on the research related to remote sensing data, this paper just uses the open source data, which does not solve the practical problems from the remote sensing and does not reflect the value of research on problems in the field of remote sensing.
Reviewer 3 Report
Comments and Suggestions for Authors
Dear authors,
The manuscript “EAD-Net: Efficiently Asymmetric Network for Semantic Labeling of High-Resolution Remote-Sensing Imagery with Dynamic Routing Mechanism” is interesting, however, it needs improvement. In the current format, it is not reproducible and easy to understand the content.
In general, I highly encourage the authors to use the IMRaD format because it will improve the readability. In the current format, it is extremely difficult to read the manuscript. There are mixed sections (Material and methods have Discussion/Results). The manuscript does not present any discussion at all, there are only results presented. A discussion section must have references to support your findings/proposals/etc. Thus, the discussion and conclusion sections could not be properly evaluated. In addition, it is required to double-check for reference styles guidelines.
Please see below some specific comments.
Title and keywords have to be improved. They cannot contain same “words”. In addition, check for typos.
Abstract: Since it was difficult to read the manuscript, the abstract could be properly evaluated. However, some typos can be found.
Line 28 – Please add the meaning of HRRSIs.
Lines 29-34 – Can the authors provide references to support the paragraph? Is there a clear period that you can distinguish the advent of “deep learning” in comparison to the “conventional image segmentation” ? What is conventional? Please clarify it.
Lines 37-40 – It is required references to support that affirmation.
Lines 53-54 – Please merge it with the next paragraph to improve readability.
Lines 59-61 – These lines are redundant with the previous lines.
Lines 69 – 83 – Please reevaluate if these lines are “contributions” or “objectives”. In the case of being “contributions”, it should be placed at the “Results and Discussion” section and therefore, the “objectives” need to be added. On the other hand, if they are “objectives” they need to be rewritten to be considered as “objectives”.
Lines 84-89 – Since it is not a “review paper”, it is not necessary to add these lines because as an “Article”, it is expected that you follow the common guidelines for “articles”. Thus, I recommend removing these lines.
Material and Methods – I highly encourage the authors to follow the IMRaD format.
Please add a paragraph introducing a Figure/Table before its appearance. In addition, please add images with higher resolution.
Lines 91-92 – “To enhance [...] named EAD-Net”. It was already stated in previous lines. There is no need to reinforce that message in the Material and Methods section if it was clearly written before.
Lines 95-112 – It confuses the reader because from the previous lines 93-94, it is expected to have the description of the four key components. Please re-organize the ideas/paragraphs to improve readability.
Lines 114 – 267 – These lines must be improved. There are material and methods mixed with discussion. It is difficult to understand the methodology.
Lines 268 – 346 – There are mixed sections.
Lines 347 – 572- “4.Experimental comparison and Analysis” – Is it supposed to be material and methods or results and discussion?
Round 2
Reviewer 1 Report
Comments and Suggestions for Authors
I've gone through the revised documents along with the previous comments and believe there are still several areas that require attention:
1. The previous comment on the second paragraph of the introduction has not been addressed. It is critical to substantiate any statements made, particularly those that assert the limitations of existing methods. I recommend reviewing recent efficient unsupervised methods, such as doi.org/10.48550/arXiv.2312.15447 and some others, which achieve quick and accurate results without requiring labels. This evidence may challenge the accuracy of the claim regarding the time-consuming and subjective nature of supervised classifiers for feature extraction in your manuscript.
2. Please ensure that throughout the document, there is a consistent space inserted between the textual content and any in-text citations. This will maintain proper formatting and readability of the manuscript.
3. The introduction should concisely summarize your proposed method and explicitly state how it tackles the challenges identified in the current literature. You could consider relocating and condensing the relevant parts of the second section to the end of the introduction.
4. In the section on evaluation metrics, it's recommended to use the standard term 'mAP' for mean average precision, assuming that is what you intend by 'mPA.' The abbreviation 'mPA' is not commonly used in this context.
5. I suggest incorporating a discussion on the choice of ResNet50 as the backbone into the ablation study section, particularly since you've already presented related results. However, the provided results focus solely on parameters and FLOPs without detailing performance metrics. It would be beneficial to also include performance metrics to give a comprehensive view of why ResNet50 was chosen, demonstrating its effectiveness in terms of not just model complexity but also accuracy or other relevant performance indicators.
6. In the conclusion, consider outlining potential future work that builds on your findings, referencing studies that indicate promising directions.
Please address all these comments.
Comments on the Quality of English LanguageMinor edits are needed.
Reviewer 3 Report
Comments and Suggestions for Authors
Despite the authors did not use the IMRaD format nor facilitate the reviewing process due to the missing highlighted changes, there are still some major issues to be addressed (i.e., mixed sections). Authors did not addressed some of the previous comments (i.e., comments 10 and 13).
Please carefully check for typos throughout the manuscript (space, comma, miss spelling words, etc). Please carefully check the writing style. There are mixing styles to present the meaning of abbreviations/acronyms (i.e., L29 and L44). In addition, if the abbreviation/acronyms has been already presented, there is no need to explain it afterwards (i.e., L51, L79 and L152). There are missing explanation of some abbreviation/acronyms (i.e., L258)
Please add proper legends to the Figures, Tables and Equations. Remember that they must be self-explanatory in the omission of its title and/or content. Therefore, if you are using abbreviations and/or acronyms and/or equation variables, please add its proper legend. The authors did not updated all the Figures/Tables accordingly to the previous comments (They must be announced in the text before its appearance – Please carefully look throughout the manuscript). Figure cannot be displayed in the middle of a text (i.e., Figure 5 and Figure 9).
Higher image resolution is required. Please note that if you look at the Figures using 100% of zoom, it is very difficult to see and clearly understand the image. The image can only be better visualized using zoom in at 290%.
L67-68 should be added with the previous paragraph.
L84 – 88, L90-101, L108-113, L138-146, L169-176, L198-203, L224-236, L325-329 they are not part of the method. They could be part of introduction or results.
Table 2 – Is that results or methods?
L358-563 – Must be improved. There are no discussion and there are “conclusions” mixed in this section (i.e., L433-438, L451-456, L470-474, L490-497). A discussion must be properly conducted comparing your results/findings with published studies.
Conclusion section cannot be evaluated since there is no discussion in the present format
Abstract can only be evaluated after changes throughout the manuscript.
